# Morphological Analysis of Laser Surface Texturing Effect on AISI 430 Stainless Steel

**DOI:** 10.3390/ma15134580

**Published:** 2022-06-29

**Authors:** Edit Roxana Moldovan, Carlos Concheso Doria, José Luis Ocaña, Bogdan Istrate, Nicanor Cimpoesu, Liana Sanda Baltes, Elena Manuela Stanciu, Catalin Croitoru, Alexandru Pascu, Corneliu Munteanu, Mircea Horia Tierean

**Affiliations:** 1Materials Engineering and Welding Department, Transilvania University of Brasov, 29 Eroilor Blvd., 500036 Brasov, Romania; edit.moldovan@unitbv.ro (E.R.M.); elena-manuela.stanciu@unitbv.ro (E.M.S.); c.croitoru@unitbv.ro (C.C.); alexandru.pascu@unitbv.ro (A.P.); 2BSH Electrodomésticos España, S.A., Avda. de la Industria 49, 50016 Zaragoza, Spain; carlos.concheso@bshg.com; 3Departamento de Física Aplicada e Ingeniería de Materiales, Universidad Politecnica de Madrid, C/José Gutiérrez Abascal 2, 28006 Madrid, Spain; joseluis.ocana@upm.es; 4Mechanical Engineering Department, Gheorghe Asachi Technical University of Iași, 43 Dimitrie Mangeron Blvd., 700050 Iași, Romania; bogdan_istrate1@yahoo.com (B.I.); cornelmun@gmail.com (C.M.); 5Materials Science Department, Gheorghe Asachi Technical University of Iași, 43 Dimitrie Mangeron Blvd., 700050 Iași, Romania; nicanornick@yahoo.com; 6Technical Sciences Academy of Romania, 26 Dacia Blvd., 030167 Bucharest, Romania

**Keywords:** ferritic stainless steel, laser surface texturing, surface patterns, morphological analysis

## Abstract

Laser surface texturing (LST) is a method to obtain micro-structures on the material’s surface for improving tribological performances, wetting tuning, surface treatment, and increasing adhesion. The material selected for LST is AISI 430 ferritic stainless steel, distinguished by the low cost in manufacturing, corrosion resistance, and high strength at elevated temperature. The present study addresses the morphology of new pattern designs (crater array, ellipse, and octagonal shapes). The patterns are applied on the stainless-steel surface by a non-contact method with high quality and precision nanosecond pulsed laser equipment. The investigation of laser parameter influence on thermal affected area and micro-structures is accomplished by morphological and elemental analysis (SEM + EDX). The parameters of the laser micro-patterning have a marked influence on the morphology, creating groove-type sections with different depths and recast material features. From the SEM characterization, the highest level of recast material is observed for concentric octagon LST design. Its application is more recommended for the preparation of the metal surface before hybrid welding. Additionally, the lack of the oxygen element in the case of this design suggests the possible use of the pattern in hybrid joining.

## 1. Introduction

Laser surface texturing (LST) is a widely used method worldwide for surface functionalization [1,2,3,4], being used in various fields: medical implants [5,6,7,8,9,10,11], wettability tuning [12], optical properties [13,14], hybrid joining [15,16,17], increasing adhesion [18], or cutting tools [19,20,21,22]. The materials that are the object of microtexturing are varied, from dentin and enamel materials, to polymers, ceramics, ferrous and non-ferrous metallic materials, and finally composite materials, in various domains such as engineering, medicine, bioengineering, etc. The most widespread LST patterns used are dimples [4,5,12,17], lines (parallel, crosshatch) [4,5,8,9,10,12,20,22], square [6,12], conical [19,22], rhombic [12,22], ripple texture (riblet texture inspired from the sharkskin, U-shapes (waviness texture) [8,18], and ring (bulge, smooth staked) [11]. The possibility of using the laser equipment to obtain texturing patterns with different geometries of high-quality surfaces to improve the materials’ performances in a single step of a technological process represents a special advantage. That is why ultrashort laser ablation is an ideal solution in creating structures with known dimensions and perfectly repeatable, on a nanoscale and beyond. Approaching this technique is a challenge, but at the same time it offers real solutions in terms of material removal.

Zhao Y. et al. [23] present Si and Cu samples laser ablated with the following characteristics of an X-ray laser: energy 50 µJ, wavelength 46.9 nm, and focused by means of a toroidal mirror at grazing incidence. Two hundred shots were applied to each of the samples, with a power density of ~2 × 10^7^ W/cm^2^. The results obtained and the related conclusion that the laser fluence is directly influenced by ablation, is consistent with the results obtained in relation with this research and presented in a previous paper of Moldovan E. et al. [24].

In value, the fluence is different, depending on the material. For example, for polymers it is between 0.1 to 1 J/cm^2^, while for inorganic insulators it is between 0.5 and 2.0 J/cm^2^ [2]. Related to the present study, Moldovan E. et al. [24] showed a decrease in fluence values from 8.49 J/cm^2^ to 2.55 J/cm^2^, in the frequency range of 30–100 kHz. The materials have a different behavior under the action of the laser. For example, the absorption of energy from a laser beam strongly influences the ablation mechanism. Thus, polymers absorb in a non-linear manner the laser energy compared to metals [2].

The literature provides information on the possibility of using nanosecond, picosecond, and femtosecond laser equipment for texturing. Pou-Álvarez P. et al. [25] studied the difference between the three variants, applying pulse lengths of 20 ns (in present study 9 ns pulse duration), 800 ps, and 266 fs on AZ31B magnesium alloy. In the nanosecond and picosecond lasers, the wavelength was 1064 nm (as in the present study) and 520 nm for the femtosecond one. The power was 6.4 W, 6.2 W, and 5.9 W (in the present study 20 W). The spot size in the nanosecond case was 163 µm (compared to 100 µm in the current study). All parameters can be analyzed comparatively in the papers [24,25,26]. SEM analysis shows uniform Mg matrix with dispersed Al-Mn-Fe particles. It is important to consider what happens due to the reduction of pulse duration. Groove profiles differ from nanosecond to femtosecond, as does the position for the recast layer. The molten material is drastically reduced with the reduction of the pulse length, the same as recast material. In the case of recast material in nanosecond situation, this is present as a uniform and thicker layer. By reducing the pulse length, an increase in width and depth of grooves is obtained.

The ablation was also studied by Cui H. et al. [27] on copper samples, using a laser with a grazing incidence angle of 7° (in the present study 3°), wavelength 46.9 nm, energy 50 µJ and 100 µJ with a pulse width of 1.7 ns. The pulse numbers were 1 to 6. By increasing the energy, the evaporation on the surface increased. At 100 µJ the average fluence were calculated at 250 mJ/cm^2^. The explanation of the mechanism was as follows: the laser energy was absorbed by the surface and after melting/evaporation the material was resolidified into nanoparticles. The placement of the sample is very important, because the vertical positioning allows, due to gravity, the nano particles to fall. The interesting results obtained encourage further research into the phenomenon.

Microtexturing also finds its place in the field of hip implants, and the couple bearing between the acetabular cup and the femoral head is very important. The combination of materials can be varied, from metal–metal to metal–ceramic and to metal–polymer. The appropriate choice of the combination of materials translates into the reduction of unwanted effects, such as aseptic loosening, tissue constraints in the body, etc. Using the finite element simulation model presented by Jamari J. et al. [28] for the combination of UHMWPE as material for acetabular cup and SS316L, CoCrMo and Ti6Al4V as metallic materials for femoral head, the contact pressure can be established for other combinations of materials.

Delgado-Ruiz R. A. et al. [5] studied zirconium dioxide for dental implants microtextured with a femtosecond laser device. They applied 120 fs pulses at 795 wavelengths, with a repetition rate of 1 kHz, a pulse energy that can reach a maximum of 1.1 mJ. As a result of surface modification, all roughness parameters significantlyincrease. Concerning the surface morphology, the pores have a conical section and the grooves a pyramidal one. The elemental analysis reveals the diminishing presence of carbon and aluminum and a larger proportion of oxygen and zirconium. The SEM analysis of surfaces showed the presence of crests and valleys and microcracks areas.

To improve osteoblastic bond of human stem cells, Cunha A. et al. [6] investigated laser surface texturing of Ti-6Al-4V alloy. The laser wavelength was 1030 nm and pulse duration 500 fs. The surface texture consisted in nanoripples, nanopillars, and microcolumns. SEM analysis shows the differences between the three types of texturing and is supplemented by X-ray photoelectron spectroscopy (XPS). Nanoripples and microcolumns induce cell stretching. Nanopillars tend to increase the formation of filopodia. After seeding and cell growth, the first two microstructures appear to be suitable for use.

In dentistry and orthodontics, it is necessary to ensure adhesion properties. Maria De la Cruz Lorenzo et al. [7] studied the microstructuring of the dentin and enamel surfaces with ultrashort pulsed laser microstructuring (wavelength: 795 nm, pulse duration 120 fs and repetition rate 1 kHz). SEM observations of the processed and failure surfaces reveal that the bonding strength is similar to other techniques and in some cases even increased.

For biomedical materials, including magnesium and titanium, laser microprocessing is beneficial. Thus, for Hu G. et al. [8], the study material was hot-extruded Mg–6Gd–0.6Ca alloy bars and Ti6Al4V. SEM images were performed on the samples after cell seeding on laser-melted and laser-melted and LIPSS technique. The wavelength of 1064 nm, the pulse width of 800 fs, and the repetition rate of 400 kHz were considered for the laser. The cell spreading was anisotropic on the laser melted and LIPSS surfaces, observing many focal adhesions (filopodia). The ossification is accelerated due to good adhesion.

Implantation of stem cells cultured in bio-resorbable polymeric scaffolds requires microstructuring technologies, useful in regenerative medicine. Poly-L-Lactide (PLLA) was tested by Rocio Ortiz et al. [9], applying grooves on the samples surfaces with a picosecond pulse laser, UV wavelength 355 nm applying an energy of 0.9 μJ at a frequency of 100 kHz, and 5 μm of pulse distance. As a result, the SEM images reveal a depression caused by material removal and protrusions of the recast material at the grooves’ ends and edges. The protrusion is the consequence of the first pulse effect. The conclusion is that grooves influence cell orientation and grooves’ edges promote cell adherence and provide guidance. The authors recommend ways to functionalize biomaterials.

Laser microprocessing using high repetition rate of a femtosecond lasers was used by Schille J. [12] for stainless steel, aluminum, and copper. SEM analysis on stainless steel, for example, reveals the hydrophobic behavior of high volumes of microcones, with contact angles of up to 150°. The reflectivity in this situation is less than 5%. The decreasing of volumes causes the abatement of hydrophobic behavior. Thermal conductivity and highly repetitive laser pulses influence the morphology of the surface materials. Around the irradiated area, a high temperature determined a higher ablation rate.

Calcium fluorite as transparent material was microstructured with a femto laser device by Rupasov A.E. et al. [13]. Nanogratings were produced at the wavelength of 515 nm and at a wavelength of 1030 nm. The doubling of the period of the nanostructures from 200–250 nm to 400–450 nm can be related with the appearance of a crater on the surface of the material showed by SEM images and the redistribution of energy in the volume.

Hybrid joints metal-polymers were of interest for Van der Straeten K. et al. [15] using X5CrNi18-10 (1.4301) stainless steel and glass fiber reinforced PA6 composite polymer. The pulse duration was about 6 ps at a wavelength of 1030 nm. SEM images revealed cone-like protrusions (CLP) which show dots and holes partially connected with bridges. The structure seems to have no predominant orientation direction. The surface roughness can be growth by increasing pulse energy and fluency.

One condition of the joint improvement is the modification of the surface roughness. Nguyen, A.T.T. et al. [16] studied metal-composite joints using adhesive, the surface being microstructured with outward/inward dimples or outward/inward grooves. It was revealed that surface irregularities ensure improved mechanical interlocking, the resin flowing through the relief being observed in SEM characterization.

Surface microtexturing has been broadly applied for improving the tribological properties of cutting tools, such as improving the friction behavior and anti-wear. The types of fabricated patterns on the cutting tool surfaces were most often micro-grooves, micro-holes, micro-stripe grooves, and micro-pools. Su Y. et al. [20] studied the influence of micro-grooved polycrystalline diamond tools manufactured with a fiber laser (scanning speed: 2 mm/s, pulse repetition rate: 20 kHz, average output power: 12 W) on Ti6Al4V, which is the workpiece material. Three cutting speeds were investigated: 16.485 m/min, 56.52 m/min, and 125.6 m/min. For the first speed the results show that the low cutting speed generates low cutting temperature and causes slight adhesion. The results for the other two speeds show excellent anti-adhesive effects compared with those of the untextured PCD tools, even without lubricants. Low tool-chip contact length can be obtained by the micro-grooved PCD tools which also determines the improvement of the friction behavior.

Nickel-based alloys have special properties, such as high corrosion resistance and high temperature resistance, but their big problem is the difficult machining. Among the materials with suitable cutting possibilities is polycrystalline cubic boron nitride (CBN), although it encounters difficulties at high cutting speeds. Sugihara T. et al. [21] chose Inconel 718 as the workpiece material and cubic boron nitride (CBN) as cutting tools, which are among the hardest materials. Grooves (grooves parallel and orthogonal to the main cutting edge, microgrooves orthogonal to the main cutting edge and placed 30 µm away from the main cutting edge) with a depth of 5 µm and width and separation of 20 µm were fabricated on the flank face of the CBN cutting tool with a femtosecond laser (wavelength 800 nm, pulse width 150 fs, cyclic frequency 1 kHz, pulse energy 300 µJ). The purpose was to stabilize the adhesion layer on the flank face and prevent it from flaking. A cutting tool with micro grooves orthogonal to the cutting edge and set back from the cutting edge significantly reduced the amount of tool retreat. This proves that microgrooving is a helpful method in machining special alloys.

The research in this article is a response to the challenge demanded by industry, namely microstructuring using nanosecond pulsed industrial laser source. Experimental research has generally shown better results in picoseconds and femtoseconds laser, but the encouraging results in the case of nanoseconds, and especially the applications for industrial and economic reasons, determined our choice in the study of the latest technical solution. The goal is to perform the morphological and elemental analysis of the laser surface texture patterns on AISI 430 stainless steel, continuing the studies completed in [24,26] for geometry, wettability, and roughness characterization.

## 2. Materials and Methods

AISI 430 ferritic stainless steel was supplied by Acerinox (ACX 500, Madrid, Spain) in the form of a large sheet, from which the samples were cut to size 80 × 25 × 0.5 mm. The material selected for the present research has a low content of carbon and nitrogen (improving the weldability, toughness, and ductility) and a good resistance in corrosive environment and exposure. Ferritic stainless steel has an elongation at room temperature (20 °C) higher than 20%, a nominal yield strength 0.2% offset greater than 260 MPa, and tensile strength between 450 MPa and 600 MPa. With low carbon and chromium between 16.00–18.00% content (Table 1), the material has a body-centered cubic (BCC) crystalline structure. After welding, the 430 ferritic stainless steel can propound corrosion at the intergranular level or/and an oxidation layer. The oxidation layer (chromium oxide) can behave as a protective layer.

The equipment used to achieve the microtexturing patterns on the surface of the ferritic stainless steel was a nanosecond pulsed laser TruMark 5020 (Trumpf Laser und Systemtechnik GmbH, Ditzingen, Germany). The laser beam is generated by an active medium Nd: Fiber Diode-pumped, with average power of 20 W, beam quality M^2^~2, 1064 nm wavelength, and 100 µm spot (at 254 mm focal distance). The 3.8 kW pulsed peak power at 20 W constant mean power ensures the highest pulse energy (>400 µJ) when frequency varies from 30 kHz to 50 kHz. The equipment has an integrated computer-aided design software (AutoCAD, version 23.0) that offers the possibility to acquire a wide variety of geometrical shapes’ design and a facile transfer between working stations. The maximum marking area is 180 × 180 mm at 254 mm focal distance and a pulse frequency from 5 kHz to 1000 kHz (9 ns pulse duration). Because of the specular reflection (surface finish of the ferritic stainless steel is bright annealed), the laser beam axis was settled at 3° angle to prevent possible damage of the laser optical fiber. Throughout the experiments, the constant parameters were: spot diameter 100 µm, power density 2.55 × 10^5^ W/cm^2^, impulses per point 1, track width of the spot 0.5 mm, overlap of the spot 99% (there is a correlation between the speed and frequency: if the speed increases/decreases the frequency increases/decreases too, to not affect the overlapping of the laser beam spot), a pulsed power of 30.5 kW at 19 kHz, and hatch distance center-to-center 0.25 mm (1st design-octagonal shape-design type A [23,24]), 0.5 mm (2nd design-two ellipses at 90° angle-type B [23,24]), and 2 mm (3rd design-crater array–type C [23,24]). The variable parameters of LST are frequency, speed, and number of repetitions. Repeatability was chosen randomly, starting from five passes, then every five repetitions up to a maximum of 15 repetitions. The maximum number of repetitions was chosen after observing the intensity of the splashes and burrs. Increasing the amount of expulsed material will increase the influenced thermal area and amount of recast material.

The laser density profile was investigated prior the laser processing to ensure that proper laser absorption into material will be obtained. The intensity distribution profile (Figure 1a) of the laser radiation on the surface plane can be described as Gaussian, which provides an image on the future groove’s shape. During the laser beam-surface interactions, a high intensity plasma surrounded by an electron charged field is formed which leads to material melting-recasting phenomena and even to delamination of the surface layer. Figure 1b shows an almost perfect outcome for the laser beam: perfect beam and real beam are almost identical.

The time cycle refers to a single sequence of repetition, decreasing with increasing frequency (Figure 2). The time cycle is an important parameter, as it can indicate the time required to perform micro-texturing. The importance comes from the influence on the capacity and volume of production that can be achieved, thus being able to influence the automation of the LST process.

The surface morphology of the samples was investigated by scanning electron microscopy (SEM FEI Quanta 200 3D Dual beam, equipped with energy-dispersive X-ray spectroscopy analysis unit-X flash Bruker, Billerica, MA, USA). The working distance is set-up at 15 mm in low vacuum mode, with a spot size 5, high voltage (20 kV), and detector LFD (Large Field Detector). Scanning electron microscopy and energy-dispersive X-ray spectroscopy provide non-destructive, rapid, qualitative, and quantitative analysis.

The microscopic analysis for sectional images and measurements (Figure 3) was performed with an optical microscope high-quality phase contrast Leica (Leica Microsystems, Heerbrugg, Switzerland, Ltd., model DMIL M LED). AutoCAD software was used to measure the ablated and recast areas.

When LST was applied on the surface of the AISI 430, a minimum of 3 mm distance was maintained from the start edge in the speed direction of each sample to mitigate the characteristic predisposition for a thermally influenced area of the ferritic stainless steel.

## 3. Results and Discussion

### 3.1. Octagonal Donuts Micro Texturing Geometry Design A

LST was applied to carve an area of 20 mm × 19.5 mm, with an edge distance of 2.75 mm in the hatch direction and 3 mm in the speed direction. The pattern applied consists of three concentric octagons, with 0.25 mm distance between contours and 1 mm twice apothem of the smallest octagon. The distance between the patterns is 0.5 mm on the hatch direction and 0.25 mm in the speed direction. For the morphological analysis, samples were selected for a low frequency and speed and an average of the number of repetitions of microtexturing (Table 2).

Draught images point out the speed direction and the positioning on the sample for LST octagonal patterns (Figure 4). A computer aided-design software (Auto-CAD) was used because it offers a wide variety of geometrical shapes, with ability to render without difficulty, and easy transfer towards processing equipment. From the top view images (Figure 5), acquired with scanning electron microscopy, one may observe a difference in size of the heat affected zone and recast material (right side is larger), caused by the angle of laser beam (3° to the left). In the right view of Figure 5, one may notice the continuous groove (successive contours), similar to the seam weld, which is due to the 99% overlapping of the laser spot. Average measurements of microtextures created with LST for pattern type A, frequency 30 kHz, 300 mm/s speed, and 10 repetitions, are 13.658 µm depth and 47.245 µm width (at the surface of the sample) with an area of 390.011 µm^2^. The recast material (the material expelled from the crevice on the edge measured in section view) has 10.479 µm height for left side and 6.735 µm for right side, with 234.194 µm^2^ area for left side and 118.017 µm^2^ for right side.

Another observation can be made regarding the splashes (Figure 6 and Figure 7). When the frequency and speed are increased, the expulsed material exceeds the edges of the groove in the form of splashes and not in the form of recast material. When the frequency and speed increase, the average values of measurements of the samples are: 66.171 µm depth, 43.012 µm width, and 1878.447 µm^2^ of area. The height of the recast material averages 25.784 µm for the left side and 26.832 µm for the right side. The area of recast material is 945.407 µm^2^ on the left side and 791.677 µm^2^ for right side of the hollow. The increase of frequency and speed outlines the growth trend of all geometrical mean measurements of the microstructures created by LST.

The SEM image of the cross section from [15] outlines the different sizes obtained for microstructure shapes as nanoripples and drops, with variations of depths (50–150 µm) and widths (10–50 µm). This variation was achieved with a picosecond pulsed laser (using 50 W laser power) by increasing the number of repetitions. The difference in the outcome after LST is obtaining regular shapes of approximately the same size (13–66 µm variations of depths and 47–43 µm of width). Similar results in terms of laser grooves, created by ablation, were also obtained in [25] in the case of nanosecond laser on magnesium samples. After the absorption of energy, the material is melted and ejected. One can thus distinguish recast material, melted and then resolidified near the ablated area. The same observation, melting, and re-solidification of the material was made by Rauh S. et al. [29] who studied nanosecond microstructuring of aluminum, with 6 ns pulse duration, 1064 nm wavelength, pulse energy 50 mJ, and laser fluence of J = 9.80 J/cm^2^ at 20 Hz repetition rate.

The EDX (Energy Dispersive X-ray) elemental analysis displays a low variation (from recast material to the deepest point of the hollow and to heat affected zone) of the elements regarding the weight and atomic percentage (Table 3). The elemental analysis shows a zero-weight percentage for oxygen, which does not apply for the ellipse and crater array patterns. The elemental analysis of octagonal microstructuring from cross-sectional images (Figure 8) shows a uniform spread outside the hollow area, excepting carbon which is also present in the cavity area. Figure 9 shows the main elements on the EDX spectrum (Cr, Fe, C).

### 3.2. Perpendicular Ellipses Micro Texturing Geometry Design B

The area marked with ellipse microtextures is 16 mm x 18 mm with an edge distance of 4.5 mm in the hatch direction and 3.63 mm in the speed direction. With a 2 mm distance center to center in both directions (hatch and speed), two ellipses are represented, overlayed at 90°, with 2 mm for length and 1 mm for width. The samples and their microtexturing parameters selected for morphological analysis are presented in Table 4.

On the draught image (Figure 10) can be highlighted the speed direction (LST direction), positioning, and number of the patterns applied. In the ellipse pattern case, one may observe more splashes and a lower deposition as recast material (Figure 11 and Figure 12). The recast material is higher on one side than the other, with the same ground as presented for previous pattern (angle of the laser beam axis).

Comparing the cross-section SEM images of designs type A (Figure 6 and Figure 7) and type B (Figure 11 and Figure 12) with those obtained by Rodríguez-Vidal E. et al. [30,31], using nanosecond fiber laser source on low alloy steel, the similar shape of the grooves is observed, having lateral recast material. In the case of the pattern design type B, a lower average of measured dimensions of the microtexture are easy to notice, compared to pattern design A. For 30 kHz frequency and 300 mm/s speed is obtained 33.203 µm depth and 54.576 µm width. The average area of the hollow is 1812.087 µm^2^. The missing recast material can be noticed in Figure 12, but the resulting height of average measurements is 20.152 µm for the left side and 23.259 µm for the right side of the edges. Compared to the previous pattern applied as LST (design type A) the design type B offers lower values of measurements, showing a differentiation in terms of applied geometry.

For ellipse pattern, a difference is the appearance of oxygen for the measurement point 2 and 3, indicating the absence of the oxygen element in the recast material and heat affected zone (Table 5). When oxygen is part of the measurement point a decreasing of iron and chromium and carbon increasing is observed.

The carbon is only spread where the cavity is (see the cross-section images of EDX analysis from Figure 13) and the spreading of the other elements (Cr, Fe, and O) is wide. Figure 14 shows the main elements on the EDX spectrum (Cr, O, Fe, C).

For type B design, a promising future is in applications in coatings processes, because of an increased area resulting from LST. This is the same as presented in [8], where tests were made on a magnesium alloy using a continuous-wave laser fiber, with gas protection (argon) and 400 kHz repetition rate of, at least, ten times higher than in the present research. The top view SEM images offer a wavy microstructure [8], with average value of periodicity of 30 µm and 14 µm for depth. The depth for pattern type B was an average of 33 µm and 55 µm for width. The presence of oxygen after LST was also highlighted in [25], where EDS analysis showed that due to the laser irradiation, an oxidation of the outer layer appeared, demonstrated by the oxygen map.

### 3.3. Crater Array Micro Texturing Geometry Design C

The marked area for crater array microtexturing pattern will be 17mm × 19.5 mm, with an edge distance of 4 mm in the hatch direction and 3.63 mm in the speed direction. The pattern is a circle/point with 0.1 mm diameter. The distance from center to center, on both directions, is 0.5 mm. For morphological analysis, samples with crater array pattern, applied with LST on the surface of a ferritic stainless steel, are presented in Table 6.

The speed direction (laser trajectory) and crater array pattern are outlined in Figure 15. For crater array pattern, a very low recast material is outlined and, in many cases, completely missing (Figure 16 and Figure 17). An increasing trend can be seen in terms of measured values for the 30 kHz frequency and 300 mm/s speed. The depth of the hollow is 127.636 µm and the width is 156.646 µm, resulting in an area of 3297.649 µm^2^.

The dimple pattern was also created in [7] using a commercial Ti: Sapphire laser with 120 fs polarized pulse at 795 nm and 1 kHz repetition rate. The measured diameter of the hole was 30 µm, but there are no data on the depth. For the type C model in this research, the width is larger (at least five times larger) due to the nanosecond pulsed laser beam, which interacts more with the surface of the material than femtosecond lasers and damages the adjacent structure and creates ripples due to the shockwave.

One can notice that the keyhole shape of the crater array shape (design type C) (Figure 17 and Figure 18) is different to that of V grooves of the designs type A and type B. This keyhole shape is specific to laser welding, described in [32,33].

The crater array model required more EDX measurement points due to the larger gaps created by the LST (Table 7). The same result as the ellipse model was observed for the carbon element (only in the cavity area). The highest percentage by weight of carbon is at the bottom of the crater (Figure 18), and the lowest percentage by weight is in the area affected by heat. Oxygen is less widespread than in previous models. Figure 19 shows the main elements on the EDX spectrum (Cr, O, Fe, C).

To assess whether there is a significant difference between the EDX elemental values for carbon and oxygen for design types B and C (i.e., if the elemental concentration values are statistically significant), Levene’s test was used, with a confidence level of 90% (significance level α = 0.1). If the p-value of Levene’s test is lower than 0.1, then obtained differences in concentration variances are unlikely to have occurred based on random sampling from a population with equal variances [34]. The design types B and C were chosen because they have similar errors in EDX elemental assessment for carbon and oxygen. These two elements were chosen for discussion due to their rather difficult quantification through the EDX method. The results of the statistical test are illustrated in Figure 20, which indicates that the p-value associated with Levene’s test is smaller than 0.1 for both carbon and oxygen, which indicates that the null hypothesis (equal random variances in elemental composition) can be rejected. Thus, micropatterning type has a statistically relevant influence on the composition of the material, especially when performed in the presence of oxygen and when being lighter and more susceptible to volatilization elements is of concern.

## 4. Conclusions

The SEM images (top view and cross section) offer information about the phenomena that occur during LST. The crater array (design type C) and octagonal (design type A) patterns present a smaller area of splashes compared to the ellipse pattern (design type B). The same phenomenon of reduction appears in the case of the recast material for ellipses and dimple/hole/crater array pattern. For pattern design type B, this is due to the 99% overlapping of the laser spot. In the case of the crater array (where there is no overlapping), the recast material is almost non-existent.

In two out of the three patterns (design type B and design type C) the presence of the oxygen element is indicated, which is beneficial for ferritic stainless steel in creating the passive layer (an oxide layer, formed from chromium and oxygen and displaying an inert reaction to the environment). The appearance of oxygen in the case of design patterns B and C, which can lead to the appearance of the passive layer (protective layer), signifies an optimal direction for the use of the patterns in tribological applications. The lack of the oxygen element in the case of pattern design type A lays out the possibility of the pattern to be used in hybrid joining.

In the area of the recast material, the EDX analysis indicates almost a double value for the carbon element (weight %) for octagonal pattern. The results, for ellipse and crater array patterns, are very appropriate (iron, chromium, and the carbon elements). At the bottom of the groove, the pattern that offers different results for elemental analysis is the octagonal design. In the thermally affected area, the measured values of the elemental analysis are appropriate for all the patterns. The cross-section images of EDX analysis offer the answer that nothing is changed regarding microtextured area.

Morphological analysis provides valuable information about the microrelief of laser textured surfaces and clues for mechanical interlocking in the case of hybrid joints. Of the three analyzed models, the highest level of recast material is observed for design type A, being more recommended in application for the preparation of the metal surface before hybrid welding.

Future studies will focus on corrosion testing, XPS, or AES characterization and FEM simulation of the microtextured specimens.

## Figures and Tables

**Figure 1 materials-15-04580-f001:**
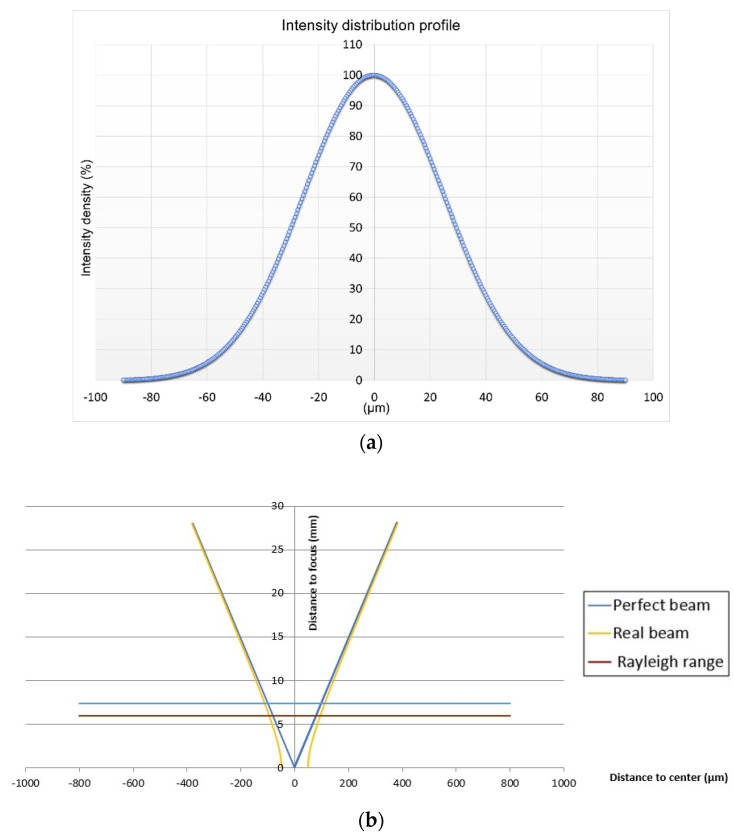
Intensity distribution profile of laser beam (**a**) and beam simulation (**b**) of TruMark 5020 laser equipment.

**Figure 2 materials-15-04580-f002:**
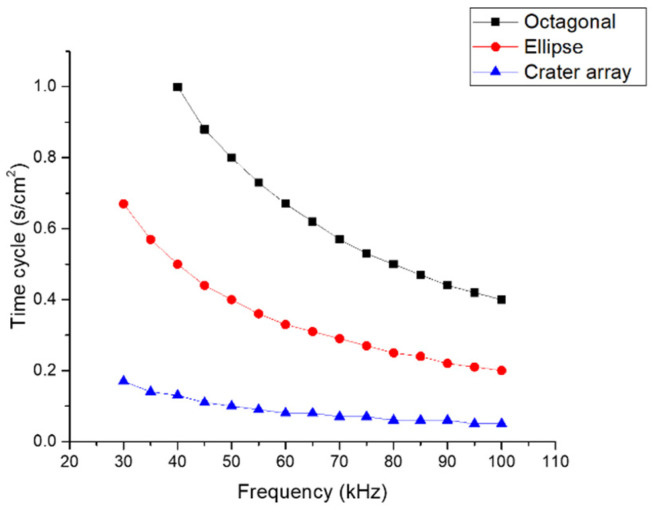
Time cycle vs. frequency.

**Figure 3 materials-15-04580-f003:**
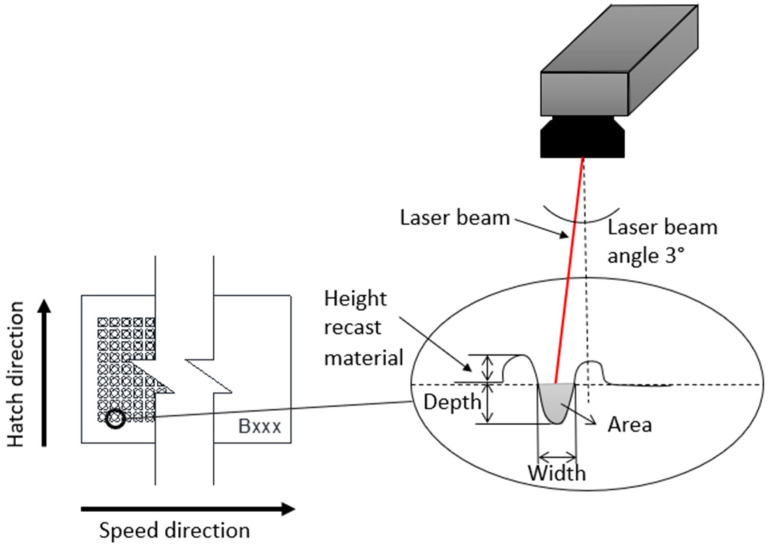
Geometry characterization after LST on ferritic stainless steel AISI 430.

**Figure 4 materials-15-04580-f004:**
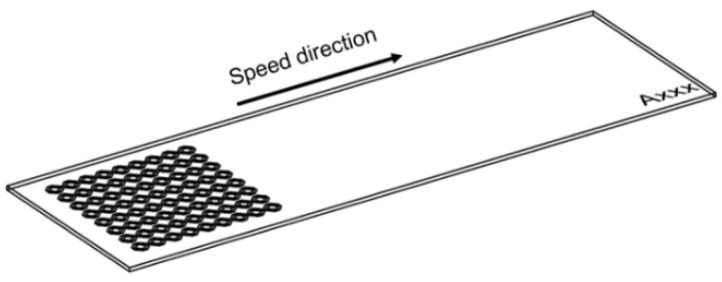
Draught image of octagonal donut shape for LST (design type A) texturing.

**Figure 5 materials-15-04580-f005:**
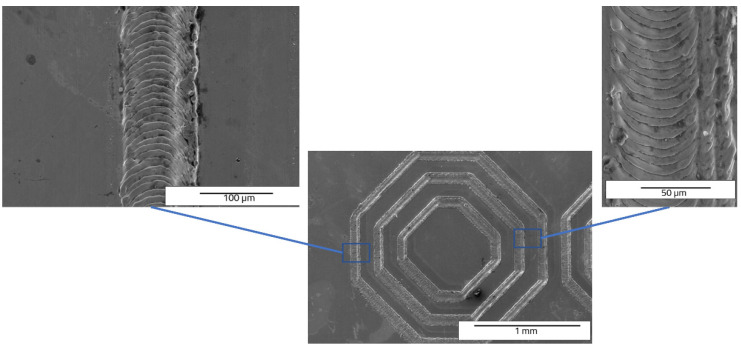
Top view SEM images of octagonal donut shape (design type A) of LST, frequency 30 kHz, speed 300 mm/s, and no. of repetitions 10.

**Figure 6 materials-15-04580-f006:**
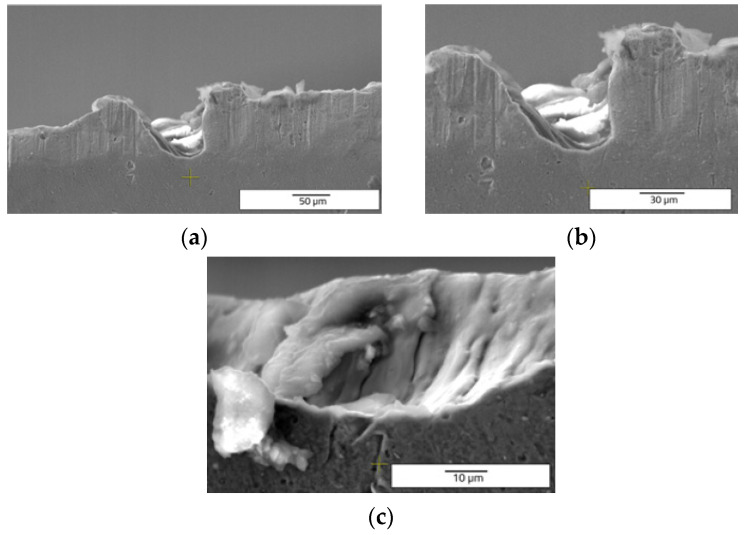
Cross section SEM images of octagonal donuts shape (design type A) of LST, frequency 30 kHz, speed 300 mm/s, and no. of repetitions 10, magnification overview of the textured groove (**a**), magnification of the LST processed region (**b**), magnification of the hollow (**c**).

**Figure 7 materials-15-04580-f007:**
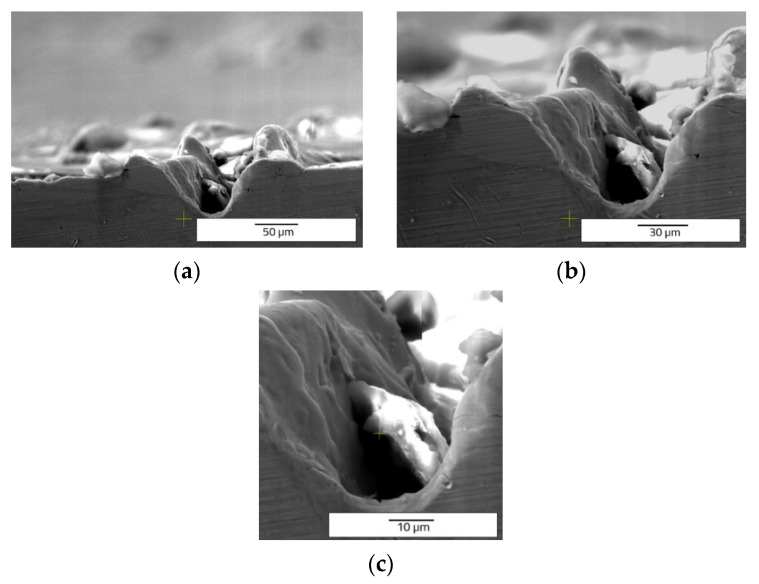
Cross section SEM images of octagonal donut shape (design type A) of LST, frequency 40 kHz, speed 400 mm/s, and no. of repetitions 10, magnification overview of the textured groove (**a**), magnification of the LST processed region (**b**), magnification of the hollow (**c**).

**Figure 8 materials-15-04580-f008:**
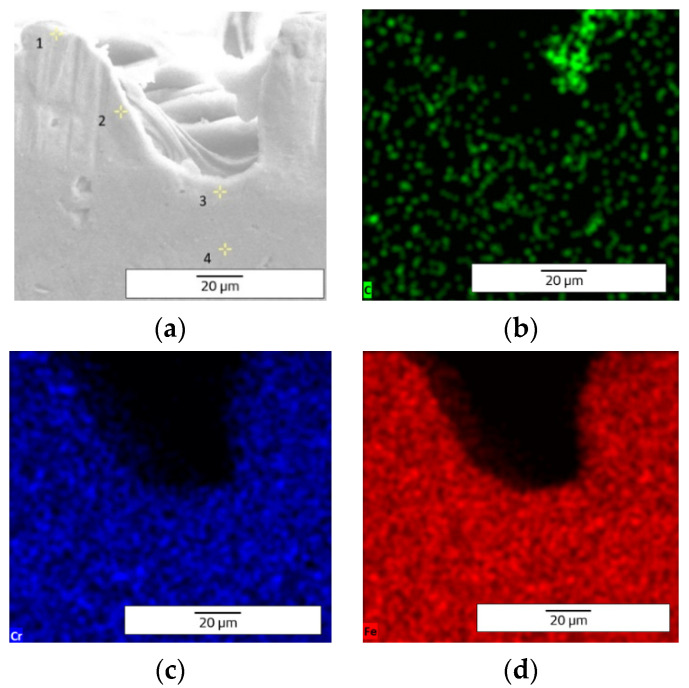
EDX elemental analysis of microtextured design octagonal donut shape (design type A) image of measurement points (**a**) and elemental mapping images (**b**—C, **c**—Cr, **d**—Fe).

**Figure 9 materials-15-04580-f009:**
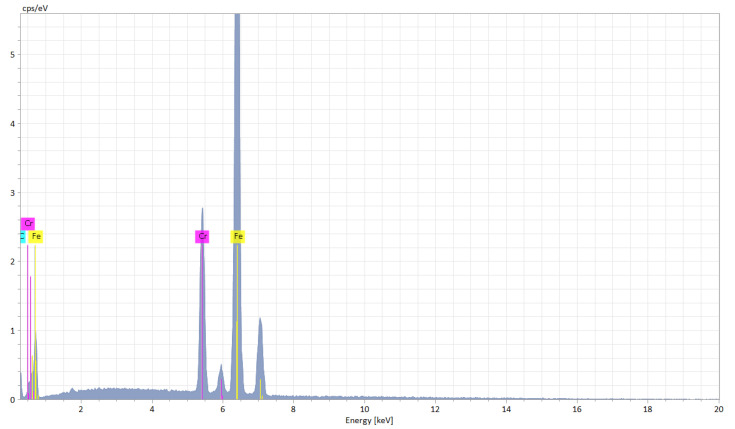
EDX spectrum of microtextured design octagonal donut shape (design type A).

**Figure 10 materials-15-04580-f010:**
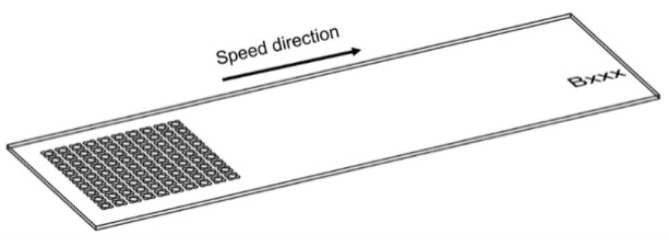
Draught image of ellipses at 90° shape for LST (design type B).

**Figure 11 materials-15-04580-f011:**
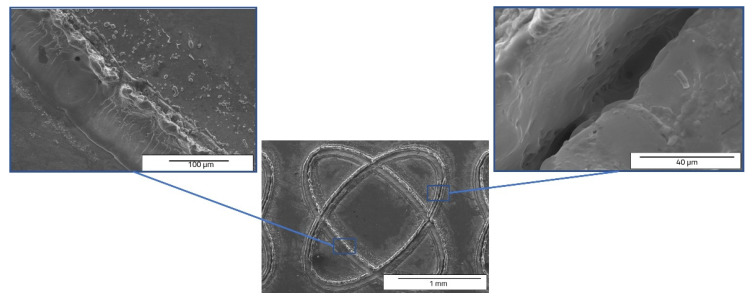
Top view SEM images of ellipses at 90° shape (design type B) of LST, frequency 30 kHz, speed 300 mm/s, and no. of repetitions 10.

**Figure 12 materials-15-04580-f012:**
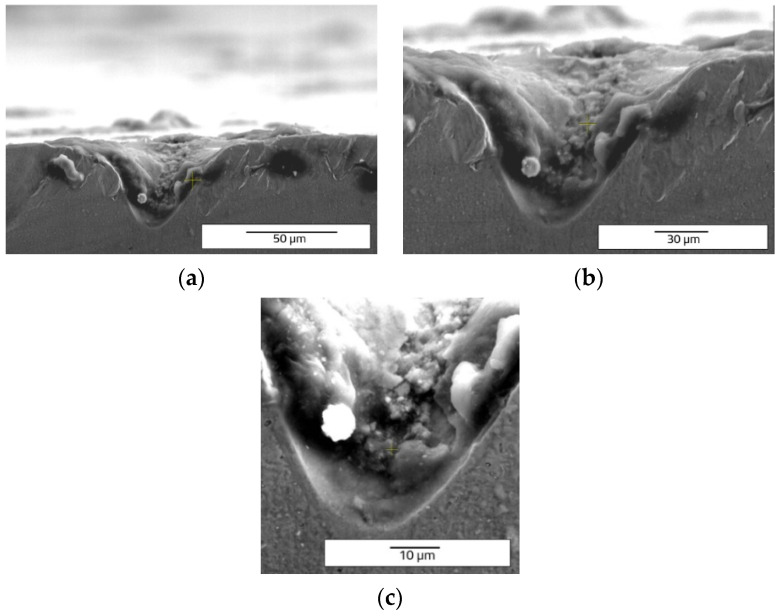
Cross section SEM images of ellipses at 90° shape (design type B) of LST, frequency 30 kHz, speed 300 mm/s, and no. of repetitions 10, magnification overview of the textured groove (**a**), magnification of the LST processed region (**b**), magnification of the hollow (**c**).

**Figure 13 materials-15-04580-f013:**
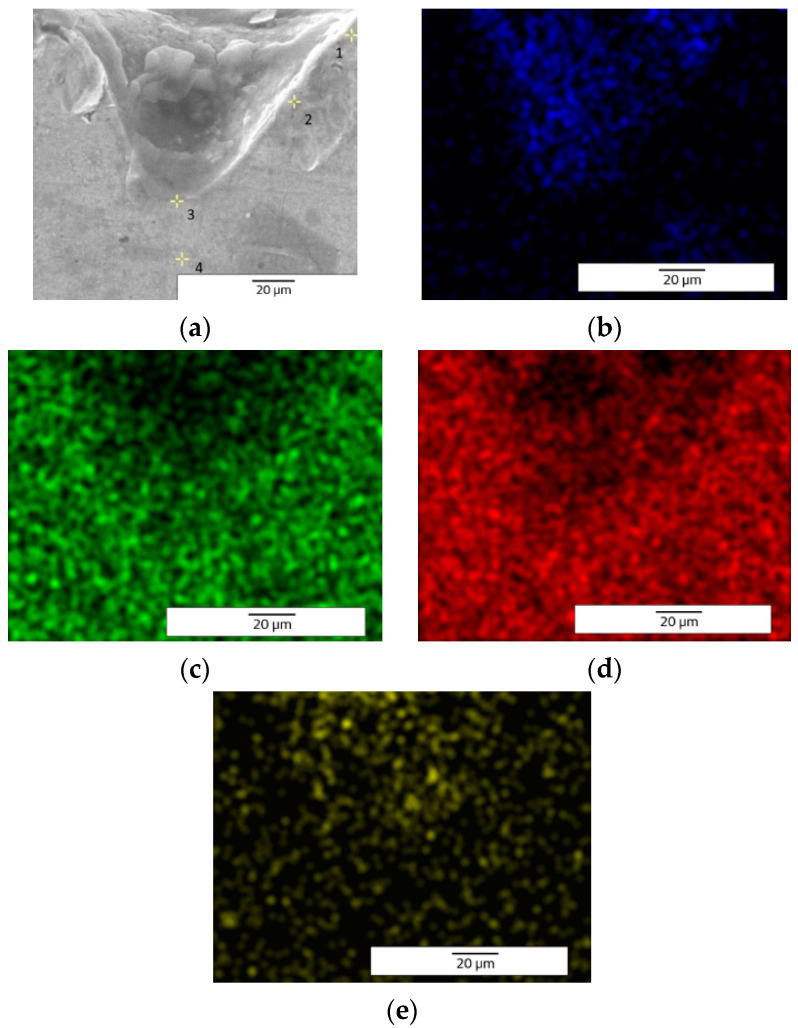
EDX elemental analysis of microtextured design ellipses at 90° shape (design type B) image of measurement points (**a**) and elemental mapping images (**b**—C, **c**—Cr, **d**—Fe, **e**—O).

**Figure 14 materials-15-04580-f014:**
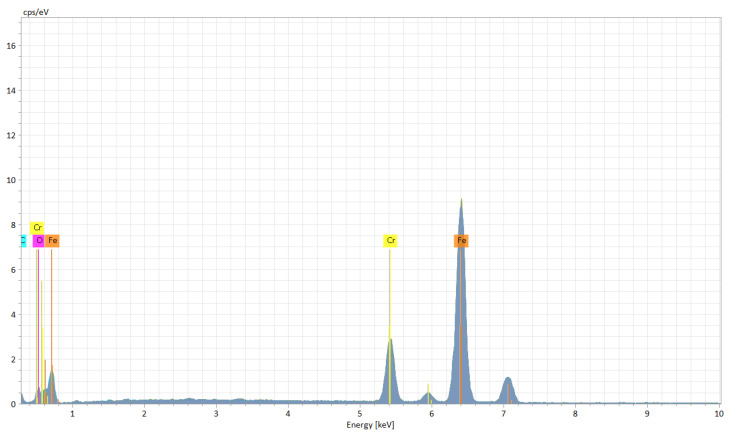
EDX spectrum of microtextured design ellipses at 90° shape (design type B).

**Figure 15 materials-15-04580-f015:**
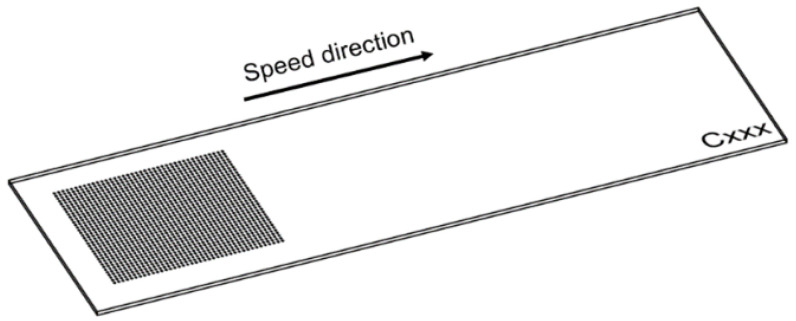
Draught image of crater array shape for LST (design type C).

**Figure 16 materials-15-04580-f016:**
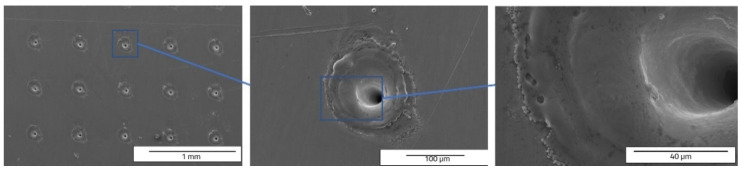
Top view SEM images of crater array shape (design type C) of LST, frequency 30 kHz, speed 300 mm/s, and no. of repetitions 10.

**Figure 17 materials-15-04580-f017:**
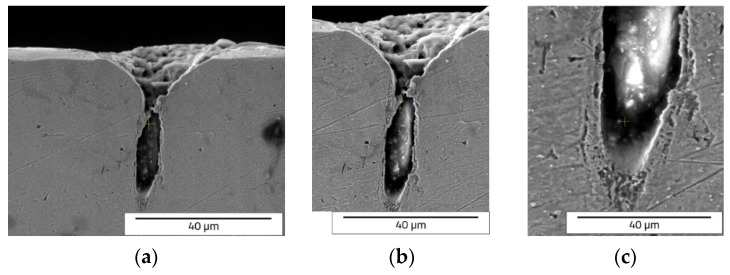
Cross section SEM images of crater array (design type C) of LST, frequency 30 kHz, speed 300 mm/s, and no. of repetitions 10, magnification overview of the textured groove (**a**), magnification of the LST processed region (**b**), magnification of the hollow (**c**).

**Figure 18 materials-15-04580-f018:**
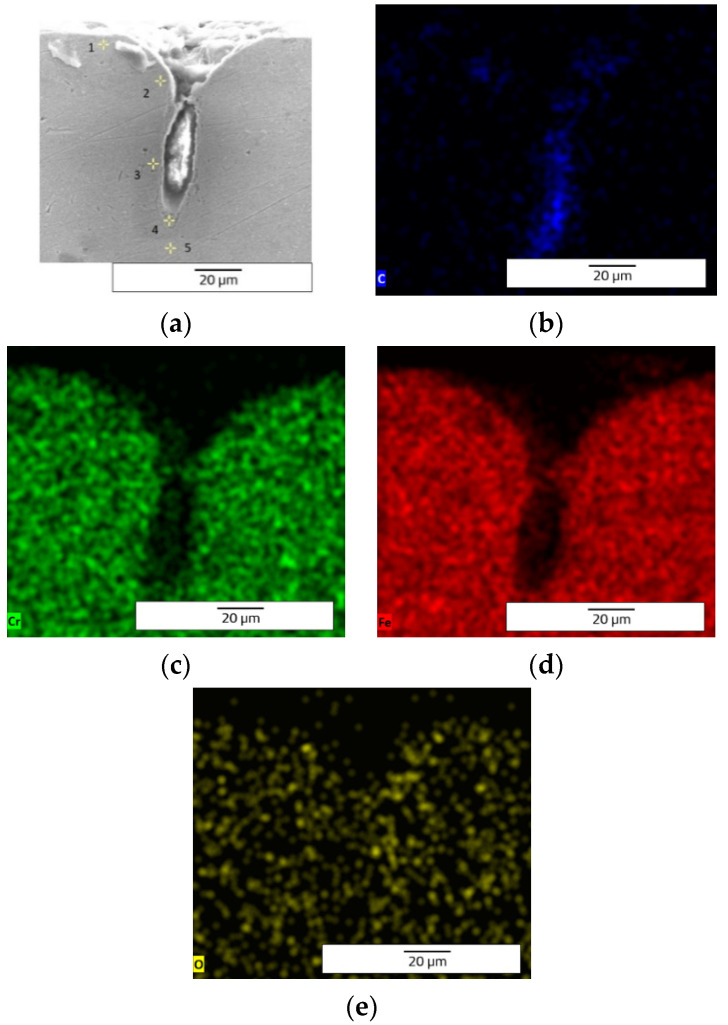
EDX elemental analysis of microtextured design crater array shape (design type C) image of measurement points (**a**) and elemental mapping images (**b**—C, **c**—Cr, **d**—Fe, **e**—O).

**Figure 19 materials-15-04580-f019:**
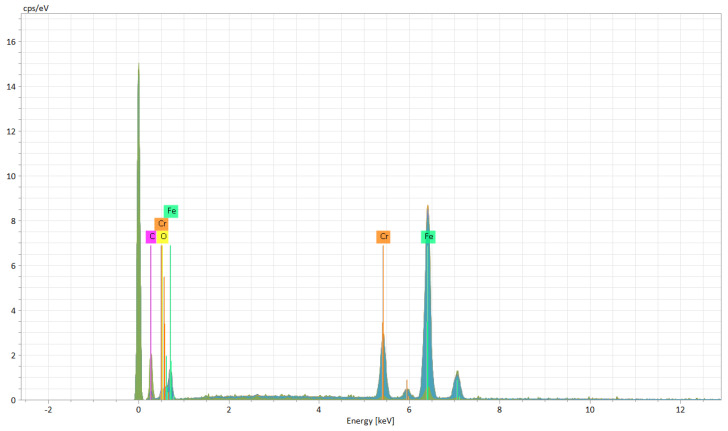
EDX spectrum of microtextured design crater array shape (design type C).

**Figure 20 materials-15-04580-f020:**
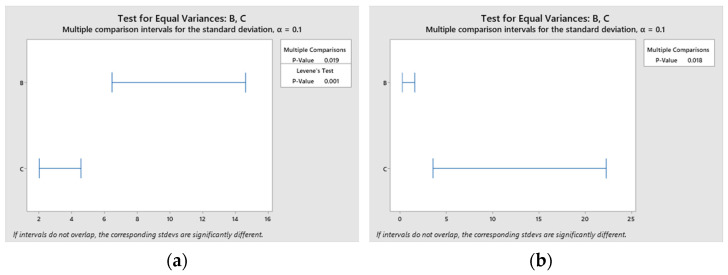
Diagrams showing Levene’s statistical test for variance significance on design types B and C: (**a**): carbon, (**b**): oxygen.

**Table 1 materials-15-04580-t001:** Chemical composition of AISI 430 stainless steel, provided by manufacturer (Acerinox, Madrid, Spain).

Chemical Element	C	Cr	N	Mn	Si	P	S
Concentration (%)	≤0.08	16.00–18.00	≤0.045	≤1.00	≤1.00	≤0.040	≤0.015

**Table 2 materials-15-04580-t002:** Variable parameters used for microtexturing design octagonal donut shape (design type A).

Sample	Frequency [kHz]	Speed [mm/s]	No. of Repetition
A3	30	300	10
A13	40	400	10

**Table 3 materials-15-04580-t003:** EDX elemental analysis of microtextured design octagonal donuts shape (design type A).

	Element	Iron		Chromium		Carbon	
Point		Weight %	Atomic %	Weight %	Atomic %	Weight %	Atomic %
1		70.58	42.78	11.85	7.72	17.57	49.5
2		72.24	47.16	13.47	9.44	14.29	43.39
3		65.55	35.72	11.82	6.92	22.64	57.36
4		70.54	44.69	13.90	9.46	15.56	45.85
Detector EDX error %		2(2.01)		0.4(2.83)		3.4(19.4)	

St. dev. (made from 5 determinations): Fe: ±0.5, Cr ± 0.1 and C ± 0.5.

**Table 4 materials-15-04580-t004:** Variable parameters used for microtexturing design ellipse shape (design type B).

Sample	Frequency [kHz]	Speed [mm/s]	No. of Repetition
B3	30	300	10
B13	40	400	10

**Table 5 materials-15-04580-t005:** EDX elemental analysis of microtextured design ellipses at 90° shape (design type B).

	Element	Iron		Chromium		Carbon		Oxygen	
Point		Weight %	Atomic %	Weight %	Atomic %	Weight %	Atomic %	Weight %	Atomic %
1		75.22	54.18	14.43	11.16	10.35	34.66	-	-	
2		57.79	28.61	12.12	6.45	22.56	51.93	7.53	13.01	
3		58.98	29.16	10.70	5.68	22.41	51.51	7.91	13.65	
4		76.52	57.35	14.62	11.76	8.86	30.88	-	-	
Detector EDX error %		1.68(2.6)		0.34(2.95)		7.98(16.52)		1.18(34.37)		

St. dev. (made from five determinations): Fe: ±0.5, Cr ± 0.1, O ± 0.2, and C ± 0.5.

**Table 6 materials-15-04580-t006:** Variable parameters used for microtexturing design crater array shape (design type C).

Sample	Frequency [kHz]	Speed [mm/s]	No. of Repetition
C3	30	300	10
C11	40	400	10

**Table 7 materials-15-04580-t007:** EDX elemental analysis of microtextured design crater array shape (design type C).

	Element	Iron		Chromium		Carbon		Oxygen	
Point		Weight %	Atomic %	Weight %	Atomic %	Weight %	Atomic %	Weight %	Atomic %
1		76.42	56.67	14.33	11.41	9.26	31.92	-	-	
2		78.41	62.50	14.92	12.77	6.67	24.73	-	-	
3		72.10	47.98	13.61	9.73	11.79	36.47	2.50	5.82	
4		59.96	30.50	11.32	6.19	20.91	49.45	7.81	13.87	
5		77.56	60.92	15.27	12.88	7.17	26.20	-	-	
Detector EDX error %		1.68(2.6)		0.34(2.95)		7.98(16.52)		1.18(34.37)		

St. dev. (made from 5 determinations): Fe: ±0.5, Cr ± 0.1, O ± 0.2, and C ± 0.5.

## Data Availability

Data sharing not applicable.

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
