# Peer review of "Morphological Analysis of Laser Surface Texturing Effect on AISI 430 Stainless Steel"

_materials, 2022, doi:10.3390/ma15134580_

Round 1

Reviewer 1 Report

Thank you for the opportunity to review this paper. This study has its merits because the authors seek to perform the morphological and elemental analysis of the laser surface texture patterns on AISI 430 stainless steel. Although the hard work in this project,  this present version is not enough to be published in the Materials journal from this reviewer's standpoint.

The Introduction section is too shallow; there is no coherence in the text; it is like a patchwork. Instead of summarizing articles, please provide a more deep basis to contrast what you are doing and the available data in the literature.

The first paragraph of the Results and Discussion should be in the M&M section.

There is a fatal flaw in this project: The capstone of the discussion is the EDX results. However, EDX is not a tool able to perform quantitative analyses. Its analytical accuracy is between ± 2% - ± 5%, and it cannot catch light elements; also,  its quantification of C and O is usually inaccurate if adventitious sources of these elements exist.  To obtain elemental % you should perform spectrometric analysis such as XPS. 

Author Response

We appreciate your constructive comments and have modified the manuscript accordingly. In the revised version the amendments and corrections made to the first submission have been highlighted. All lines indicated below are referred to the Track changes.doc file.

The Introduction section is too shallow; there is no coherence in the text; it is like a patchwork. Instead of summarizing articles, please provide a more deep basis to contrast what you are doing and the available data in the literature.

We updated whole Introduction according to your recommendations (lines 44-102, 104-112, 207-211).

The first paragraph of the Results and Discussion should be in the M&M section.

The first paragraph of the Results and Discussion was moved in the end of M&M section (lines 282-284).

There is a fatal flaw in this project: The capstone of the discussion is the EDX results. However, EDX is not a tool able to perform quantitative analyses. Its analytical accuracy is between ± 2% - ± 5%, and it cannot catch light elements; also, its quantification of C and O is usually inaccurate if adventitious sources of these elements exist.  To obtain elemental % you should perform spectrometric analysis such as XPS.

We are aware of the limitations of the EDX detector anyway we are using the quantitative results to compare the areas and procedures and the error involve in determination of C or O will be the same in each case. The distributions of the elements are in agreement with the quantitative results from the tables and even the C percentages are suitable for high detector errors, the values obtained are appreciate and compared between them and nor with other results from other articles. We insert in Table 3 and Table 5 the detector absolute error and in parentheses the relative error, both given by EDX detector. The conclusions made on EDX results are only qualitative and slightly quantitative. Further research on the field will include the XPS or AES determinations. Tables 3,5,7 were modified:

Table 3. EDX elemental analysis of micro textured design octagonal donuts shape (design type A).

Element

Point

Iron

Chromium

Carbon

weight %

atomic %

weight %

atomic %

weight %

atomic %

1

70.58

42.78

11.85

7.72

17.57

49.5

2

72.24

47.16

13.47

9.44

14.29

43.39

3

65.55

35.72

11.82

6.92

22.64

57.36

4

70.54

44.69

13.90

9.46

15.56

45.85

Detector EDX error %

2(2.01)

0.4(2.83)

3.4(19.4)

St. dev. (made from 5 determinations): Fe:±0.5, Cr±0.1 and C±0.5.

Table 5. EDX elemental analysis of micro textured design ellipses at 90° shape (design type B).

Element

Point

Iron

Chromium

Carbon

Oxygen

weight %

atomic %

weight %

atomic %

weight %

atomic %

weight %

atomic %

1

75.22

54.18

14.43

11.16

10.35

34.66

-

-

2

57.79

28.61

12.12

6.45

22.56

51.93

7.53

13.01

3

58.98

29.16

10.70

5.68

22.41

51.51

7.91

13.65

4

76.52

57.35

14.62

11.76

8.86

30.88

-

-

Detector EDX error %

1.68(2.6)

0.34(2.95)

7.98(16.52)

1.18(34.37)

St. dev. (made from 5 determinations): Fe:±0.5, Cr±0.1, O±0.2 and C±0.5.

Table 7. EDX elemental analysis of micro textured design crater array shape (design type C).

Element

Point

Iron

Chromium

Carbon

Oxygen

weight %

atomic %

weight %

atomic %

weight %

atomic %

weight %

atomic %

1

76.42

56.67

14.33

11.41

9.26

31.92

-

-

2

78.41

62.50

14.92

12.77

6.67

24.73

-

-

3

72.10

47.98

13.61

9.73

11.79

36.47

2.50

5.82

4

59.96

30.50

11.32

6.19

20.91

49.45

7.81

13.87

5

77.56

60.92

15.27

12.88

7.17

26.20

-

-

Detector EDX error %

1.68(2.6)

0.34(2.95)

7.98(16.52)

1.18(34.37)

St. dev. (made from 5 determinations): Fe:±0.5, Cr±0.1, O±0.2 and C±0.5.

Lines 469-485 were introduced:

To assess if there is a significant difference between the EDX elemental values for carbon and oxygen for design types B and C (i.e., if the elemental concentration values are statistically significant), Levene's test was used, with a confidence level of 90% (significance level α=0.1). If the p-value of Levene’s test is lower than 0.1, then obtained differences in concentration variances are unlikely to have occurred based on random sampling from a population with equal variances [34].  The design types B and C were chosen because they have similar errors in EDX elemental assessment for carbon and oxygen. These two elements were chosen into discussion due to their rather difficult quantification through the EDX method. The results of the statistical test are illustrated in Figure 20, which indicates that p-value associated with Levene’s test is smaller than 0.1 for both carbon and oxygen, which indicates that the null hypothesis (equal random variances in elemental composition) can be rejected. Thus, micropatterning type has a statistically relevant influence on the composition of the material, especially when performed in the presence of oxygen and when lighter and more susceptible to volatilization elements are of concern.

(a)

(b)

Figure 20. Diagrams showing Levene’s statistical test for variance significance on design types B and C: (a): carbon, (b): oxygen

Lines 516-517 were introduced:

Future studies will focus on corrosion testing, XPS or AES characterization and FEM simulation of the microtextured specimens.

Reviewer 2 Report

The MS presents the surface texturing of stainless steel with nanosecond pulsed lasers. It is well-organzied and the experiments are comprehensive and thorough. It will be better that FIg.1b is rotated 90degress by convention, which will be more easily understood. 

Author Response

We appreciate your constructive comments and have modified the manuscript accordingly.

  1. It will be better that Fig.1b is rotated 90degress by convention, which will be more easily understood. 

The present positioning of Figure 1b is that during the experiments because the laser beam has a vertical axis, being transmitted from top to bottom, where is the workpiece, located at the end of the focusing distance (vertical focusing). In Figure 1b, for clarity, we changed the color for the Rayleigh range.

Reviewer 3 Report

The article titled: “Morphological Analysis of Laser Surface Texturing Effect on AISI 430 Stainless Steel”, where authors have reported the interaction of the laser pulse and matters. Morphological Analysis of Laser Surface Texturing Effect on AISI 430 Stainless Steel is presented. However, there are some major points needed to be explained further before publication, as follows;

1.      Please add two references in the introduction portion with the explanation related to this work, as follows:

[1] Optics Express, Vol. 23 ( Title: Si and Cu ablation with a 46.9-nm laser focused by                     a toroidal mirror, page 14126-14134) (2015).
[2] Appl. Sci. 2020, 10, 214; doi:10.3390/app10010214 (Title: Study of Thermal Effect in the Interaction of Nanosecond Capillary Discharge Extreme Ultraviolet Laser with Copper)

2.      The simulation is based on a heat transfer model solving differential equations, which is suitable for technical problems, while here the photon energy involved and the solid physics demand a more detailed physical treatment, including quantum effects. If you perform a heat transfer treatment will only find such effects.

3.      The author should give the wavelength and energy of the laser used for the irradiation of target.

4.      Please show the detail process to calculate the temperature. Maybe, the authors should show the equation set for temperature calculation with some assumptions.

5.      The experimental part is a collection of micrographs without any real breakthrough, and missing a control/refernce variable. A study should explore a parameter by step changing its value and investigating the outcomes. A statistical treatment is required to validate the robustness of the findings. Further I miss at all reference data, e.g. images before and after the irradiation, to be able to make a case.

6.      The experimental part however is lacking results of systematic studies: What is the fluence dependence of the damage? What is the influence of the pulse number on damage (way more than showing in images)?

Author Response

We appreciate your constructive comments and have modified the manuscript accordingly. In the revised version the amendments and corrections made to the first submission have been highlighted. All lines indicated below are referred to the Track changes.doc file.

  1. Please add two references in the introduction portion with the explanation related to this work, as follows:

[1] Optics Express, Vol. 23 ( Title: Si and Cu ablation with a 46.9-nm laser focused by toroidal mirror, page 14126-14134) (2015).
[2] Appl. Sci. 2020, 10, 214; doi:10.3390/app10010214 (Title: Study of Thermal Effect in the Interaction of Nanosecond Capillary Discharge Extreme Ultraviolet Laser with Copper).

We introduced the recommended papers in the references as:

  1. Zhao, Y; Cui, H; Zhang, W; Li, W; Jiang, S; Li, L. Si and Cu ablation with a 46.9-nm laser focused by a toroidal mirror. Opt Express. 2015, 23(11), 14126-34.
  2. Cui, H.; Zhao, Y.; Khan, M.U.; Zhao, D.; Fan, Z. Study of Thermal Effect in the Interaction of Nanosecond Capillary Discharge Extreme Ultraviolet Laser with Copper. Appl. Sci. 2020, 10, 214.

and we updated the introduction accordingly in lines 57-62:

Zhao Y. et al [23] present Si and Cu samples laser ablated with the following characteristics of an X-ray laser: energy 50µJ, wavelength 46.9 nm and focused by means of a toroidal mirror at grazing incidence. 200 shots were applied to each of the samples, with a power density of ~2 × 107 W/cm2. The results obtained and the related conclusion that the laser fluence is directly influenced by ablation, is consistent with the results obtained in relation with this research and presented in previous paper of Moldovan E. et al. [24].

and 85-93:

The ablation was also studied by Cui H. et. al. [27] on copper samples, using a laser with a grazing incidence angle of 7° (in the present study 3°), wavelength 46.9 nm, energy 50µJ and 100 µJ with a pulse width of 1.7 ns. The pulse numbers were 1 to 6. Increasing the energy, the evaporation on the surface accentuated. At 100 µJ the average fluence were calculated at 250 mJ/cm2. The explanation of the mechanism was as follows: the laser energy was absorbed by the surface and after melting / evaporation the material was resolidified into nanoparticles. The placement of the sample is very important, because the vertical positioning allows, due to gravity, the nano particles to fall. The interesting results obtained encourage further research deep into the phenomenon.

  1. The simulation is based on a heat transfer model solving differential equations, which is suitable for technical problems, while here the photon energy involved and the solid physics demand a more detailed physical treatment, including quantum effects. If you perform a heat transfer treatment will only find such effects.

This is a full experimental work. We introduced in lines 516-517:

Future studies will focus on corrosion testing, XPS or AES characterization and FEM simulation of the microtextured specimens.

  1. The author should give the wavelength and energy of the laser used for the irradiation of target.

We introduced the lines 233-235:

1064 nm wavelength and 100 µm spot (at 254 mm focal distance). The 3.8 kW pulsed peak power at 20 W constant mean power, ensue the highest pulse energy (>400 µJ) when frequency varies from 30 kHz to 50 kHz.

  1. Please show the detail process to calculate the temperature. Maybe, the authors should show the equation set for temperature calculation with some assumptions.

Because this paper is based on optical, SEM and EDX characterization we don’t calculate the temperature.

  1. The experimental part is a collection of micrographs without any real breakthrough, and missing a control/refernce variable. A study should explore a parameter by step changing its value and investigating the outcomes. A statistical treatment is required to validate the robustness of the findings. Further I miss at all reference data, e.g. images before and after the irradiation, to be able to make a case.

The goal of this work was explained in lines 206-213. We introduced the lines 207-211:

Experimental research has generally shown better results in picoseconds and femtoseconds laser, but the encouraging results in the case of nanoseconds and especially the impositions for industrial economic reasons, determined our choice in the study of the latest technical solution.

  1. The experimental part however is lacking results of systematic studies: What is the fluence dependence of the damage? What is the influence of the pulse number on damage (way more than showing in images)?

We introduced the lines 309-314:

Average measurements of microtextures created with LST for pattern type A, frequency 30 kHz, 300 mm/s speed and 10 repetitions, are 13.658 µm depth and 47.245 µm width (at the surface of the sample) with an area of 390.011 µm2. The recast material (the material expelled from the crevice on the edge) measured in section view points out a height of 10.479 µm for left side and 6.735 µm for right side, with an area of 234.194 µm2 for left side and 118.017 µm2 right side.

the lines 327-333:

When the frequency and speed increase, the average values of measurements of the samples are: 66.171 µm depth, 43.012 µm width and 1878.447 µm2 of area. The height of the recast material averages 25.784 µm for the left side and 26.832 µm for the right side. The area of recast material, on left side is 945.407 µm2 and 791.677 µm2 for right side of the hollow. The increase of frequency and speed outlines the growth trend of all geometrical mean measurements of the microstructures created by LST.

the lines 394-402:

In the case of the pattern design type B, a lower average of measured dimensions of the micro texture are easy to notice, compared to pattern design A. For 30 kHz frequency and 300 mm/s speed is obtained 33.203 µm depth and 54.576 µm width. The average area of the hollow is 1812.087 µm2. The missing recast material can be noticed in Figure 12, but the resulted height of average measurements is 20.152 µm for the left side and 23.259 µm for the right side of the edges. Comparing to the previous pattern applied as LST (design type A) the design type B is offering lower values of measurements, showing a differentiation in terms of applied geometry.

and the lines 435-438:

For crater array pattern, a very low recast material is outlined and, in many cases, completely missing (Figures 16,17). An increasing trend can be seen in terms of measured values for the 30 kHz frequency and 300 mm/s speed.  The depth of the hollow is 127.636 µm and the width is 156.646 µm, resulting an area of 3297.649 µm2.

Reviewer 4 Report

1.      Qualitative results need to be added in the abstract section.

2.      Reorder keywords in ablatival order.

3.      The reviewer does not see any something really new in the present study, it has been similar previous study. The authors need to highlight their novelty.

4.      State of the art and research gap should be explained clearly understood in the introduction section.

5.      The procedure of experimental texting needs to be illustrated as the form of a figure in the materials and methods section, provide it.

6.      Experimental standard needs to be stated.

7.      Tools specification and detailed procedure needs to be explained in the introduction section.

8.      Please make a description of the figure/table with its figure/table it continues in the same page, not a separated page.

9.      For the figure, please arrange it to make fit into one page. Further typesetting is needed.

10.   Since the Laser Surface Texturing brings benefits to enhancing the performance of metallic hip implants, the suggested reference published by MDPI needs to be adopted as follows: Computational Contact Pressure Prediction of CoCrMo, SS 316L and Ti6Al4V Femoral Head against UHMWPE Acetabular Cup under Gait Cycle. J. Funct. Biomater. 2022, 13, 64. https://doi.org/10.3390/jfb13020064

11.   Write a paragraph at least 3 sentences to make it not too short and give a solid explanation, the authors made the paragraphs only consist of one or two sentences int the present article, please revise it.

12.   Results need to be compared with another similar study if possible.

13.   The discussion needs to be extended. My recommendation is analysis in the perspective of finite element analysis besides on experimental testing. My previous suggested reference may be adopted.

14.   Please recheck the English used and revise its error.

15.   Limitations of the conducted study should be mentioned.

16.   Collusion is not concise, please rewrite it.

17.   Further studies should be explained in the conclusion section.

18.   Overall, the manuscript is lack of quality. Serious improvement is needed. Especially in the introduction section.

Author Response

We appreciate your constructive comments and have modified the manuscript accordingly. In the revised version the amendments and corrections made to the first submission have been highlighted. All lines indicated below are referred to the Track changes.doc file.

  1. Qualitative results need to be added in the abstract section.

We introduced in abstract section the lines 31-34

From the SEM characterization, the highest level of recast material is observed for concentric octagons LST design. Its application is more recommended for the preparation of the metal surface before hybrid welding. Also, the lack of the oxygen element in the case of this design, lays out the possibility of the pattern to be used in hybrid joining.

  1. Reorder keywords in ablatival order.

We changed the keywords in lines 37-38

ferritic stainless steel, laser surface texturing, surface patterns, morphological analysis

  1. The reviewer does not see any something really new in the present study, it has been similar previous study. The authors need to highlight their novelty.

We introduced the lines 207-211 in Introduction:

Experimental research has generally shown better results in picoseconds and femtoseconds laser, but the encouraging results in the case of nanoseconds and especially the impositions for industrial economic reasons, determined our choice in the study of the latest technical solution.

  1. State of the art and research gap should be explained clearly understood in the introduction section.

We updated whole Introduction according to your recommendations (lines 44-102, 104-112, 207-211).

  1. The procedure of experimental texting needs to be illustrated as the form of a figure in the materials and methods section, provide it.

Figure 3 was introduced to illustrate the experimental procedure:

Figure 3. Geometry characterization after LST on ferritic stainless steel AISI 430. 

  1. Experimental standard needs to be stated.

The SEM and EDX characterization procedures are those provided by the equipment manufacturers.

  1. Tools specification and detailed procedure needs to be explained in the introduction section.

We introduced the lines 276-279:

The microscopic analysis for sectional images and measurements (Figure 3) were performed with an optical microscope high-quality phase contrast Leica (Leica Microsystems, Switzerland, Ltd, model DMIL M LED). AutoCAD software was used to measure the ablated and recast areas.

The rest of equipment and software used are detailed in chapter 2 Materials and Methods.

  1. Please make a description of the figure/table with its figure/table it continues in the same page, not a separated page.

Whole text and figures have been reorganized as requested.

  1. For the figure, please arrange it to make fit into one page. Further typesetting is needed.

We hope to fit the figures in one page in the proofreading version of this paper.

  1. Since the Laser Surface Texturing brings benefits to enhancing the performance of metallic hip implants, the suggested reference published by MDPI needs to be adopted as follows: Computational Contact Pressure Prediction of CoCrMo, SS 316L and Ti6Al4V Femoral Head against UHMWPE Acetabular Cup under Gait Cycle. J. Funct. Biomater. 2022, 13, 64. https://doi.org/10.3390/jfb13020064

We introduced the recommended paper in the references as:

  1. Jamari, J.; Ammarullah, M.I.; Santoso, G.; Sugiharto, S.; Supriyono, T.; Prakoso, A.T.; Basri, H.; van der Heide, E. Computational Contact Pressure Prediction of CoCrMo, SS 316L and Ti6Al4V Femoral Head against UHMWPE Acetabular Cup under Gait Cycle. J. Funct. Biomater. 2022, 13, 64.

and we updated the introduction accordingly in lines 94-102:

Microtexturing also finds its place in the field of hip implants, the couple bearing between the acetabular cup and the femoral head is very important. The combination of materials can be varied, from metal-metal to metal-ceramic and to metal-polymer. The inspired choice of the combination of materials translates into the reduction of unwanted effects, such as aseptic loosening, tissue constraints in the body, etc. Using the finite element simulation model presented by Jamari J. et al. [28] for the combination of UHMWPE as material for acetabular cup and SS316L, CoCrMo and Ti6Al4V as metallic materials for femoral head, the contact pressure can be established for other combinations of materials.

  1. Write a paragraph at least 3 sentences to make it not too short and give a solid explanation, the authors made the paragraphs only consist of one or two sentences int the present article, please revise it.

Whole text has been reorganized as requested.

  1. Results need to be compared with another similar study if possible.

We introduced the lines 336-348:

The SEM image of the cross section from [15] is outlining the different sizes obtained for microstructures shapes as nanoripples and drops, with variations of depths (50-150 µm) and widths (10-50 µm). This variation was achieved with a picosecond pulsed laser (using 50W laser power) by increasing the number of repetitions. The difference of the resulted outcome after LST, is offering regular shapes with approximately the same size (13-66 µm variations of depths and 47-43 µm of width). Similar results in terms of laser grooves, created by ablation, were also obtained in [25] in the case of nanosecond laser on magnesium samples. After the absorption of energy, the material is melted and ejected. One can thus distinguish recast material, melted, and then resolidified near the ablated area. The same observation, melting and re-solidification of the material, was made by Rauh S. et al [29] who studied nanosecond microstructuring of aluminum, with 6 ns pulse duration, 1064 nm wavelength, pulse energy 50 mJ, laser fluence of J=9.80 J/cm2 at 20 Hz repetition rate.

We introduced the lines 417-425:

For type B design, a promising future is for applications in coatings processes, because of an increased area resulted after LST, same as presented in [8], where tests were made on magnesium alloy using a continuous-wave laser fiber, with gas protection (argon) and 400 kHz repetition rate of, at least ten times higher than in the present research. The top view SEM images are offering a waviness microstructure [8], with average value of periodicity of 30 µm and 14 µm for depth. The depth for pattern type B was an average of 33 µm and 55 µm for width. The presence of oxygen after LST was also highlighted in [25], where EDS analysis showed that due to the laser irradiation, an oxidation of the outer layer appeared, demonstrated by the oxygen map.

We introduced the lines 444-449:

The dimple pattern was also created in [7] using a commercial Ti: Sapphire laser with 120 fs polarized pulse at 795 nm and 1 kHz repetition rate. The measured diameter of the hole was 30 µm, but there is no data on the depth. For the type C model in this research, the width is larger (at least five times larger) due to the nanosecond pulsed laser beam, which interacts more with the surface of the material than femtosecond lasers and damages the adjacent structure and creates ripples due to the shockwave.

  1. The discussion needs to be extended. My recommendation is analysis in the perspective of finite element analysis besides on experimental testing. My previous suggested reference may be adopted.

FEM simulation will be the next step of our research. We introduced accordingly the last paragraph of Conclusions.

  1. Please recheck the English used and revise its error.

English language was rechecked.

  1. Limitations of the conducted study should be mentioned.

In tables 3,5,7 was introduced the EDX detector error:

Table 3. EDX elemental analysis of micro textured design octagonal donuts shape (design type A).

Element

Point

Iron

Chromium

Carbon

weight %

atomic %

weight %

atomic %

weight %

atomic %

1

70.58

42.78

11.85

7.72

17.57

49.5

2

72.24

47.16

13.47

9.44

14.29

43.39

3

65.55

35.72

11.82

6.92

22.64

57.36

4

70.54

44.69

13.90

9.46

15.56

45.85

Detector EDX error %

2(2.01)

0.4(2.83)

3.4(19.4)

St. dev. (made from 5 determinations): Fe:±0.5, Cr±0.1 and C±0.5.

Table 5. EDX elemental analysis of micro textured design ellipses at 90° shape (design type B).

Element

Point

Iron

Chromium

Carbon

Oxygen

weight %

atomic %

weight %

atomic %

weight %

atomic %

weight %

atomic %

1

75.22

54.18

14.43

11.16

10.35

34.66

-

-

2

57.79

28.61

12.12

6.45

22.56

51.93

7.53

13.01

3

58.98

29.16

10.70

5.68

22.41

51.51

7.91

13.65

4

76.52

57.35

14.62

11.76

8.86

30.88

-

-

Detector EDX error %

1.68(2.6)

0.34(2.95)

7.98(16.52)

1.18(34.37)

St. dev. (made from 5 determinations): Fe:±0.5, Cr±0.1, O±0.2 and C±0.5.

Table 7. EDX elemental analysis of micro textured design crater array shape (design type C).

Element

Point

Iron

Chromium

Carbon

Oxygen

weight %

atomic %

weight %

atomic %

weight %

atomic %

weight %

atomic %

1

76.42

56.67

14.33

11.41

9.26

31.92

-

-

2

78.41

62.50

14.92

12.77

6.67

24.73

-

-

3

72.10

47.98

13.61

9.73

11.79

36.47

2.50

5.82

4

59.96

30.50

11.32

6.19

20.91

49.45

7.81

13.87

5

77.56

60.92

15.27

12.88

7.17

26.20

-

-

Detector EDX error %

1.68(2.6)

0.34(2.95)

7.98(16.52)

1.18(34.37)

St. dev. (made from 5 determinations): Fe:±0.5, Cr±0.1, O±0.2 and C±0.5.

Lines 469-485 were introduced:

To assess if there is a significant difference between the EDX elemental values for carbon and oxygen for design types B and C (i.e., if the elemental concentration values are statistically significant), Levene's test was used, with a confidence level of 90% (significance level α=0.1). If the p-value of Levene’s test is lower than 0.1, then obtained differences in concentration variances are unlikely to have occurred based on random sampling from a population with equal variances [34].  The design types B and C were chosen because they have similar errors in EDX elemental assessment for carbon and oxygen. These two elements were chosen into discussion due to their rather difficult quantification through the EDX method. The results of the statistical test are illustrated in Figure 20, which indicates that p-value associated with Levene’s test is smaller than 0.1 for both carbon and oxygen, which indicates that the null hypothesis (equal random variances in elemental composition) can be rejected. Thus, micropatterning type has a statistically relevant influence on the composition of the material, especially when performed in the presence of oxygen and when lighter and more susceptible to volatilization elements are of concern.

(a)

(b)

Figure 20. Diagrams showing Levene’s statistical test for variance significance on design types B and C: (a): carbon, (b): oxygen

  1. Collusion is not concise, please rewrite it.

We updated the conclusions: lines 490-492

The same phenomenon of reduction appears in the case of the recast material, for ellipses and dimple/hole/crater array pattern. For pattern design type B is due to the 99% overlapping of the laser spot.

lines 499-503:

The appearance of oxygen in the case of design patterns B and C, which can lead to the appearance of the passive layer (protective layer), signifies an optimal direction for the use of the patterns in tribological applications. The lack of the oxygen element in the case of pattern design type A, lays out the possibility of the pattern to be used in hybrid joining.

  1. Further studies should be explained in the conclusion section.

We introduced in conclusions: lines 516-517:

Future studies will focus on corrosion testing, XPS or AES characterization and FEM simulation of the microtextured specimens.

  1. Overall, the manuscript is lack of quality. Serious improvement is needed. Especially in the introduction section.

We updated the introduction to be more consistent.

Round 2

Reviewer 1 Report

The authors responded adequately to the points from the first review.

Reviewer 3 Report

Authors have revised the manuscript very well and addressed all the points carefully. Manuscript is written and compiled very well. I would like to see the article publish in the present form.

Reviewer 4 Report

It has been improved, well done.